:ۥ: PLOS | ONE

# Diversity, distribution and dynamics of large trees across an old-growth lowland tropical rain forest landscape

David B. Clark○[1]*, Antonio Ferraz[2], Deborah A. Clark[1], James R. Kellner[3,4], Susan G. Letcher○[5], Sassan Saatchi[2]

**1** Department of Biology, University of Missouri-St. Louis, St. Louis, Missouri, United States of America, **2** NASA Jet Propulsion Laboratory, California Institute of Technology, Pasadena, California, United States of America, **3** Institute at Brown for Environment and Society, Brown University, Providence, Rhode Island, United States of America, **4** Department of Ecology and Evolutionary Biology, Brown University, Providence, Rhode Island, United States of America, **5** Plant Biology, College of the Atlantic, Bar Harbor, Maine, United States of America

* dbclark50@yahoo.com

**Data Availability Statement:** All relevant data are within the manuscript and its Supporting Information files.

## Abstract

Large trees, here defined as ≥60 cm trunk diameter, are the most massive organisms in tropical rain forest, and are important in forest structure, dynamics and carbon cycling. The status of large trees in tropical forest is unclear, with both increasing and decreasing trends reported. We sampled across an old-growth tropical rain forest landscape at the La Selva Biological Station in Costa Rica to study the distribution and performance of large trees and their contribution to forest structure and dynamics. We censused all large trees in 238 0.50 ha plots, and also identified and measured all stems ≥10 cm diameter in 18 0.50 ha plots annually for 20 years (1997–2017). We assessed abundance, species diversity, and crown conditions of large trees in relation to soil type and topography, measured the contribution of large trees to stand structure, productivity, and dynamics, and analyzed the decadal population trends of large trees. Large trees accounted for 2.5% of stems and ~25% of mean basal area and Estimated Above-Ground Biomass, and produced ~10% of the estimated wood production. Crown exposure increased with stem diameter but predictability was low. Large tree density was about twice as high on more-fertile flat sites compared to less fertile sites on slopes and plateaus. Density of large trees increased 27% over the study interval, but the increase was restricted to the flat more-fertile sites. Mortality and recruitment differed between large trees and smaller stems, and strongly suggested that large tree density was affected by past climatic disturbances such as large El Niño events. Our results generally do not support the hypothesis of increasing biomass and turnover rates in tropical forest. We suggest that additional landscape-scale studies of large trees are needed to determine the generality of disturbance legacies in tropical forest study sites.

**Funding:** Funding was provided by U.S. National Science Foundation grants 1357177 (DBC, DAC), 1357097 (JRK), and 1147367 (DAC, SGL), and by U.S. National Aeronautics and Space Agency (NASA) Terrestrial Ecology grant TE08-0037 (SS, AF, DBC). This work was partially carried out at the Jet Propulsion Laboratory, California Institute of Technology, and University of California Los Angeles, under a contract by NASA (16-ESUSPI-16-0015) (SS). The funders had no role in study design, data collection and analysis, decision to publish, or preparation of the manuscript.

**Competing interests:** The authors have declared that no competing interests exist.

## Introduction

The most massive organisms of tropical rain forests are the largest trees. Large trees in tropical forests have attracted increasing attention due to their role in carbon cycling, their contributions to forest structure and dynamics [1,2,3,4], and their reported sensitivity to droughts and forest fragmentation [5,6,7,8,9].

Another interest in large trees is their use as proxies for whole-forest variables such as Estimated-Above-Ground Biomass (EAGB) [2,10,11,12,13]. Several studies have shown that measurements of large tree crown size and height based on remotely-sensed data can be generalized to forest attributes over large spatial domains, cf [12,14,15].

There are two seemingly contradictory views of population trends of large trees in tropical rain forests. There have been a number of reports that pantropically forests are accumulating biomass and turning over faster (the Bigger and Faster Hypothesis [16,17, 18]). If large trees are behaving as the average trees in these forests, then large trees should be increasing in size and exhibiting faster turnover. In contrast, several studies have found greater sensitivity of large trees to drought than for smaller stems [6, 7, 8, 19], and large trees have been reported to be in decline in many areas [20,21]. Large trees may also be considerably more sensitive than smaller trees to death from lighting and invasive pests [22]

Evaluating these contrasting views with ground data in tropical forests has been difficult because of the challenge of obtaining statistically powerful samples of these large trees. Because the density of large trees per hectare is relatively low (~10–30 per ha in a broad range of tropical forests [23]), many hectares have to be sampled to accumulate a useful sample. Taking into account the rarity of most individual tropical tree species [24], it is also not surprising that little is known about the species-level ecology of most large trees (exceptions include [15,25,26,27]). Ground-based studies have frequently been based on one or a few plots, and thus intra-landscape variation in large tree density and performance has rarely been quantified from ground data. Because large trees are both large and frequently buttressed, it is challenging to make accurate repeated diameter measurements, and the active discussion of large tree ontogenetic growth patterns is in part due to these sampling and measurement issues [28,29].

Recent advances in remote sensing have led to progress in assessing both the landscape-scale distribution of large trees [26,27,30] and the performance of individuals over a several-year interval [26, 27, 31]. To date however no study has attempted to quantify current large tree distribution, diversity and demography over an old-growth tropical rain forest landscape, and to link current-day distribution with recent decadal trends in tree performance and local disturbance.

In the research reported here we had three main goals:

1. To assess the abundance, species diversity, and individual crown conditions of large trees across an old-growth tropical rain forest landscape in relation to soil type and topography

2. To measure the contribution of large trees to stand structure, productivity and dynamics, and

3. To determine the decadal population trends of large trees over this landscape.

We addressed the challenge of large tree rarity by inventorying large trees in 238 0.50-ha plots sited across an upland old-growth tropical rain forest landscape with well-described variation in soil nutrients and topography [32,33,34]. To evaluate decadal trends in density of large trees, we drew on data from 18 0.50 ha forest inventory plots where all stems $\geq$10 cm diameter were measured annually for 20 years. Both data sets were used to analyze variation in density and performance in relation to edaphic gradients.

The definition of a large tree is inherently comparative (bigger vs. smaller) and subjective (how big?). Different research groups have used different criteria, in part depending on whether the research was based on ground-based or remotely-sensed data (cf ≥70 cm [35,36,37], ≥60 cm, [23], ≥ 50 cm [14]). The research reported here was based on ground data, and we used the criterion of a trunk diameter ≥60 cm in order to compare our data to the largest possible number of tropical studies. We publish here all the original data from this research so that future studies can use these data with any large tree classification that may prove useful.

In this study we found that large tree density was increasing over this old-growth landscape, consistent with the Bigger and Faster Hypothesis. However, we also found clear demographic signals that the increasing density of large trees was most likely due to recovery from the effects of the historically-unusual high frequency of large El Niño events over the last four decades, and that overall, stand dynamism has decreased over the last two decades. We conclude by examining the possibility of landscape-scale climatic disturbance legacies on tropical forest inventory plots globally.

## Materials and methods

This research was conducted under permits from the Costa Rican Ministerio de Ambiente y Energia, most recently Resolution SINAC-ACC-PI-R-037-2018. This study was carried out at the La Selva Biological Station in N.W. Costa Rica. The forest is classified as Tropical Wet Forest in the Holdridge Life Zone system [38]. All the areas sampled were terra firme (upland) forest and are considered old-growth forest [39]

We analyzed two tree inventory data sets. One came from a long-term study of forest dynamics (the CARBONO Project) based on annual censuses of all stems >10 cm diameter in 18 0.50-ha (50 x 100 m) plots. The plots were established in three edaphic conditions [33] using a stratified random design: flat sites on old alluvial soils, flat sites on residual soils, and steeply-sloping plots on residual soils (Fig 1). Complete details of plot establishment and project protocols are available at the CARBONO web page of the Organization for Tropical Studies (https://tropicalstudies.org/). We used these data to calculate growth, death, and recruitment rates of large trees and all smaller stems, as well as to analyze biodiversity and distribution patterns. Estimated above-ground biomass (EAGB) was calculated using Brown's [40] tropical wet forest allometry, which incorporates fewer assumptions than estimates including wood density and gives similar values at this site [41]. All the CARBONO data analyzed here were originally collected by D.B. Clark and D.A. Clark. All data necessary to replicate the CARBONO plot analyses in this paper are provided the Supplementary Information (S1 and S2 Tables). We refer to this dataset as the "long-term inventory" sample.

A second data set in this study consisted of complete inventory of all large trees within 238 0.50-ha plots (50 x 100 m) in five blocks distributed across the mesoscale old-growth landscape at La Selva (Fig 1). The permanent monuments of the La Selva 50 x 100 m grid system were used as plot corners. For the field work plot boundaries were delineated with fabric meter tapes and all large trees were censused in each plot. Diameters were measured with a fabric diameter tape (+/- 1 mm) at 1.3 m height or using up to 6 m of ladder for individuals with buttresses or basal swellings. For 5 of the 1622 individuals a good measurement site was not available within ~7 m of the ground; for these individuals, diameter was estimated by comparison to an extended m tape held perpendicular to the trunk at ~7 m. Crown position on a 7-point scale [25] was estimated for each individual by two different observers, and these estimates were averaged. The 238 large tree census plots were established over a 6-year interval (2006–2011) and were last censused for mortality in 2016. Death rates for the individuals added each

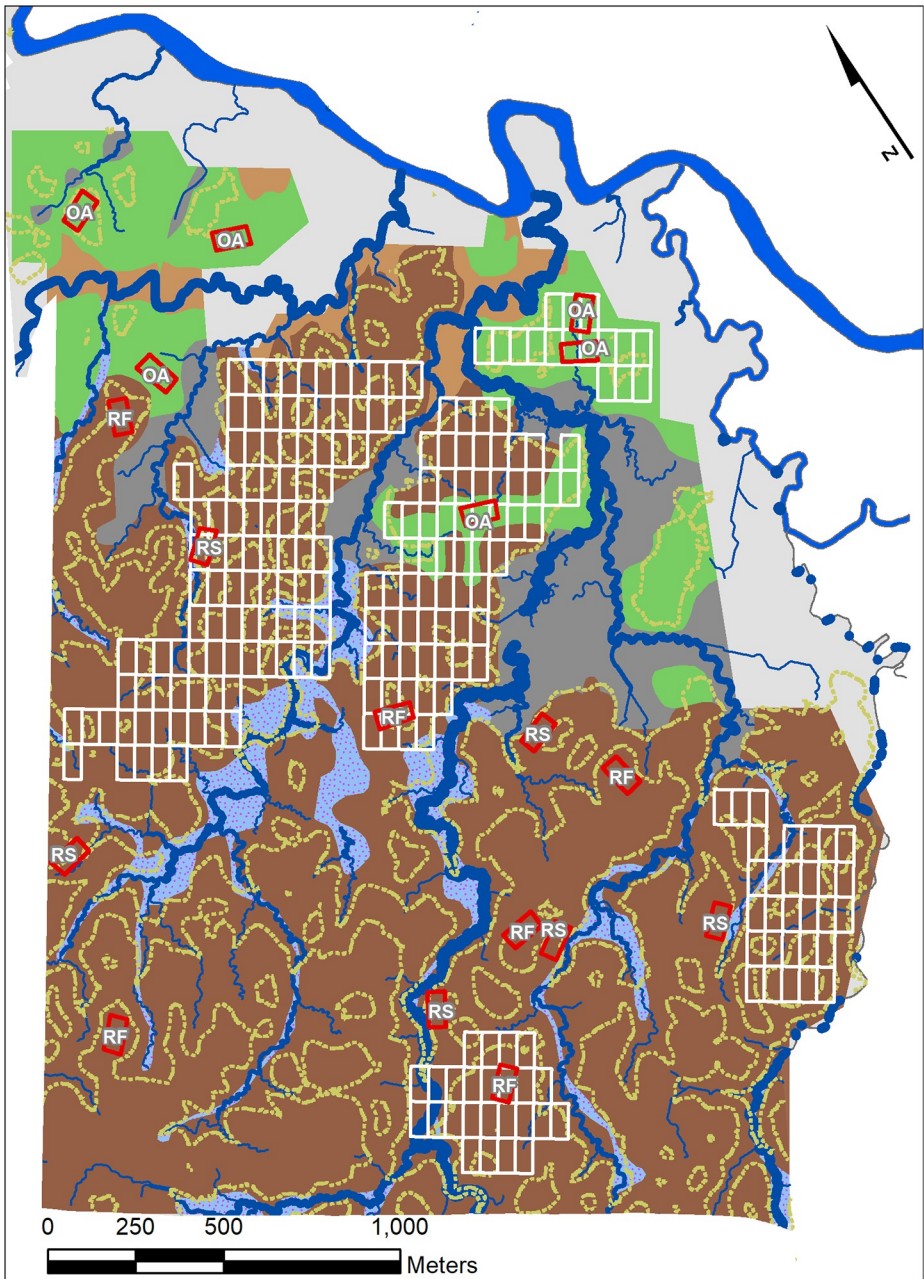

**Fig 1. Study area.** Soil types and study areas in old-growth tropical rain forest at the La Selva Biological Station, Costa Rica. Soil types [33] are residual (dark brown), old alluvial (green), swamp (gray), riparian (light gray stippled) and recent alluvium (light brown). 18 0.5 ha long-term inventory plots are show as red rectangles (RF = residual soil flat sites, RS = steeply-sloping residual soil, OA = flat old alluvial sites). White rectangles show the 238 0.50 ha plots with complete inventories of large trees.

year were calculated to 2016. The data for these large trees are provided in Supporting Information (S3 Table). We refer to these plots as the "large tree plots".

A long history of soils research at La Selva has led to the classification of the major soil types. The dominant upland soils are flat upland soils of alluvial origin near the Puerto Viejo River, and older residual soils derived from in-place weathering of basaltic lava flows with

increasing elevation up the stream watersheds ([32] Fig 1). The alluvial soils in general have higher nutrient concentrations than the residual soils (S4 Table) [42,43]. The long-term inventory plots generally support the view of more nutrient rich old alluvial soils, although erosion rejuvenates nutrient supply on residual slopes [44]. Concentrations of P and K were higher and C/N ratios lower in the top 10 cm of soil in plots located on old alluvium compared to the flat and steeply-sloping residual soils plots. Floristic and structural differences between sites on residual and old alluvial soils are well documented [16,33,45,46].

We used lidar data to develop a digital elevation model (DEM) and characterize slopes in the long-term inventory and large tree plots. The lidar data was acquired in 2009 by the Northrop Grumman Corporation using an Optech ALTM 3100 scanning device. We used a comprehensive point cloud averaging 6 points per square meter that was filtered to identify ground and vegetation points [47]. We interpolated a DEM (1 m spatial resolution) from a Delaunay Triangulation that was defined as a function of the lidar ground points. A slope angle map was calculated by estimating the maximum rate of change in elevation from a given grid cell to its eight neighbors.

## Results

### Relative importance of large trees

In the long-term inventory plots large trees accounted for an average 2.5% of stems from 1997–2017, roughly one quarter of mean basal area and EAGB, and about 10% of estimated annual increments in basal area and EAGB over this period (Table 1).

Mean abundance and productivity of large trees and smaller stems ($\geq$ 10 cm diameter to <60 cm) in 18 0.50 ha long-term inventory plots across an old-growth upland tropical rain forest landscape at La Selva Biological Station, Costa Rica, 1997–2017. Data for structure variables are based on means of 21 annual censuses per 0.50 ha plot (N = 18) from 1997–2017. Data for annual productivity are based on mean plot-level values for 20 annual recensuses from 1998–2017. SEM = standard error of the mean, EAGB = Estimated Above-Ground Biomass estimated using Brown's Tropical Wet Forest allometry [40].

We calculated the 20-year mean EAGB for all stems in each plot and separately for the large trees and smaller stems. Variation in total plot EAGB largely explained by variation in large tree EAGB per plot ($r^2_{(adj)}$ = 0.69, P<0.001, N = 18). However the percentage of total plot EAGB represented by large trees increased significantly with plot EAGB ($r2_{(adj)}$ = 0.56, P<0.001, N = 18), so the statistical significance of the relation of total plot EAGB to large tree EAGB is likely to be affected by the part-whole correlation issue. Total plot EAGB of smaller stems was negatively correlated with plot large tree EAGB ($r^2_{(adj)}$ = 0.17, P<0.06).

### Large tree species diversity and size

For a point-in-time assessment of large tree species diversity we analyzed the 2009 census data from the long-term inventory plots (18 0.5 ha plots). A total of 218 tree species were recorded

**Table 1. Stem abundance and productivity.**

| | Large trees | | Smaller stems | | Total | % Large trees |
|---|---|---|---|---|---|---|
| | Mean | SEM | Mean | SEM | | |
| Stems/plot | 6.2 | 0.8 | 239.4 | 9.5 | 245.6 | 2.5 |
| Basal area/plot m2 | 2.61 | 0.36 | 9.31 | 0.25 | 11.92 | 21.9 |
| EAGB/plot Mg | 21.7 | 3.0 | 60.2 | 1.6 | 81.8 | 26.5 |
| Annual basal area increment increment/plot (m$^2$) | 0.028 | 0.005 | 0.258 | 0.011 | 0.286 | 9.8 |
| Annual EAGB increment/plot Mg | 0.25 | 0.05 | 1.94 | 0.07 | 2.19 | 11.3 |

among the ≥10 cm diameter stems; of these 218 species 25 (11.5%) had at least one individual reaching the 60 cm large tree size limit. The 25 species were distributed among 15 families and 24 genera.

For a multidecadal view of large tree species diversity we analyzed the annual diameter measurements of all stems ≥10 cm diameter in the long-term inventory plots from 1997–2017. There were 67,153 diameter measurements on 4,635 individual trees (excluding palms and lianas). Over these two decades 241 species of trees occurred in the plots. Of the 241 species, 32 had at least one diameter measurement of ≥60 cm ("large tree species") and 209 did not. Rarity and small size were correlated. The 78 rarest species were all non-large tree species with an average maximum diameter of 18.7 cm. Overall the large tree species tended to be more common (median 19 individuals in the total sample versus 5 for non-large tree species) and larger than the non-large tree species (mean diameter 34.8 vs. 17.1 cm).

In the 238 0.5 ha large tree plots, the 1662 large trees occurred in 34 families, 60 genera, and 70 species (Table 2). *Pentaclethra macroloba* (Fabaceae) was the dominant species, accounting for 51% of large tree individuals. Fabaceae was the dominant large tree family with 13 genera and 18 species, and accounted for 65% of all large tree individuals. The second-most abundant family (Meliaceae), with two species in separate genera, accounted for only 6% of individuals. The number of large tree species decreased as stem diameter increased (Table 3), and only 18 species reached 100 cm diameter.

For individual large trees that were mapped by the GIS analysis to either residual or old alluvial soils (N = 1552), 12 species accounted for 76% of the total large tree sample. The frequency distribution of 12 species (and all others grouped as one) was highly significantly different between the two soils types (Pearson chi-square 42.2, df = 12, P<0.001), indicating strong edaphic preferences of the large tree species.

## Large tree crown exposure to light

As expected for the largest trees in the forest, most large trees had crowns that were either fully exposed to vertical illumination or were emergent with both vertical and full lateral exposure to light (Table 4). Large trees with crowns in these high-light conditions accounted for 80% of the total sample of large trees and 82% of the total large tree basal area in this sample (Table 4). There was a general tendency for the larger-diameter trees to have more highly illuminated crowns (Fig 2), but the low degree of predictability is notable. Large trees with emergent crowns occurred across the entire size diameter size range, and many large-diameter individuals were not emergent (Fig 2). The lack of a high correlation between diameter and crown position was probably related to several factors including specific topographic microsite, local neighborhood forest structure, an individual's history of stem and crown damage, and morphological differences among species. For example *Pentaclethra macroloba*, the most common large tree species (51% of the landscape sample), had a significantly lower distribution of crown positions than the non-Pentaclethra large trees (Table 4, $\chi^2$ df = 3, P<0.0001).

## Distribution of large trees in relation to topography

The experimental design of the CARBONO Project, with plots stratified across local gradients of soil type and topography, facilitated analyses edaphic effects on the distribution of large trees and smaller stems (Table 5). In plots located on residual soils, there were no detectable effects of flat versus steep-slope topography on any aspect of stand structure (Table 5 column Slope Effect).

We also analyzed the effects of topography on large tree density in the long-term inventory plots using GIS-derived estimates of mean plot elevation and slope. A two-factor model

**Table 2. Size and abundance of large tree species.**

| Rank | % | N | Genus | Species | Family | Diameter (cm) | | | | |
|---|---|---|---|---|---|---|---|---|---|---|
| | | | | | | Mean | Min | Max | SEM | Q05 |
| 1 | 51.2 | 831 | Pentaclethra | macroloba | Fabaceae | 71.1 | 60.0 | 133.0 | 0.4 | 90.6 |
| 2 | 4.6 | 74 | Balizia | elegans | Fabaceae | 89.4 | 60.1 | 149.6 | 2.4 | 125.4 |
| 3 | 3.9 | 63 | Carapa | nicaraguensis | Meliaceae | 85.8 | 60.1 | 134.2 | 2.3 | 124.7 |
| 4 | 2.8 | 45 | Virola | koschnyi | Myristicaceae | 70.8 | 60.0 | 90.8 | 1.3 | 86.9 |
| 5 | 2.7 | 43 | Vitex | cooperi | Lamiaceae | 77.9 | 60.0 | 108.8 | 2.0 | 106.3 |
| 6 | 2.5 | 41 | Laetia | procera | Salicaceae | 68.6 | 60.0 | 86.6 | 0.9 | 79.6 |
| 7 | 2.5 | 40 | Apeiba | membranacea | Malvaceae | 78.7 | 60.1 | 118.1 | 2.4 | 116.6 |
| 8 | 2.4 | 39 | Guarea | guidonia | Meliaceae | 71.5 | 60.3 | 100.9 | 1.5 | 92.9 |
| 9 | 2.0 | 32 | Stryphnodendron | microstachyum | Fabaceae | 66.8 | 60.0 | 76.9 | 0.8 | 76.3 |
| 10 | 1.8 | 29 | Vochysia | ferruginea | Vochysiaceae | 74.2 | 60.0 | 91.0 | 1.7 | 91.0 |
| 11 | 1.8 | 29 | Dipteryx | panamensis | Fabaceae | 92.1 | 65.0 | 146.0 | 3.6 | 139.4 |
| 12 | 1.6 | 26 | Inga | alba | Fabaceae | 67.4 | 60.0 | 86.3 | 1.7 | 85.6 |
| 13 | 1.5 | 24 | Ilex | skutchii | Aquifoliaceae | 86.2 | 63.0 | 107.4 | 2.2 | 106.3 |
| 14 | 1.5 | 24 | Hieronyma | alchorneoides | Phyllanthaceae | 101.9 | 60.2 | 161.0 | 6.0 | 157.5 |
| 15 | 1.2 | 20 | Lecythis | ampla | Lecythidaceae | 90.3 | 60.2 | 132.0 | 4.6 | 131.6 |
| 16 | 1.1 | 18 | Tapirira | guianensis | Anacardiaceae | 66.5 | 60.0 | 85.4 | 1.5 | 85.4 |
| 17 | 1.1 | 18 | Dussia | macroprophyllata | Fabaceae | 70.2 | 60.6 | 88.4 | 2.0 | 88.4 |
| 18 | 1.1 | 18 | Terminalia | amazonia | Combretaceae | 83.0 | 61.4 | 136.9 | 4.9 | 136.9 |
| 19 | 1.0 | 17 | Sacoglottis | trichogyna | Humeriaceae | 78.1 | 62.8 | 94.0 | 2.4 | 94.0 |
| 20 | 1.0 | 17 | Tachigali | costaricensis | Fabaceae | 82.4 | 62.7 | 117.7 | 3.8 | 117.7 |
| 21 | 0.9 | 15 | Minquartia | guianensis | Coulaceae | 65.2 | 60.1 | 73.0 | 1.2 | 73.0 |
| 22 | 0.9 | 14 | Hernandia | didymantha | Hernandiaceae | 68.6 | 60.2 | 76.4 | 1.3 | 76.4 |
| 23 | 0.8 | 13 | Conceveiba | pleiostemona | Euphorbiaceae | 75.2 | 64.3 | 97.5 | 2.6 | 97.5 |
| 24 | 0.8 | 13 | Alchorneopsis | floribunda | Euphorbiaceae | 70.4 | 60.1 | 109.7 | 3.6 | 109.7 |
| 25 | 0.8 | 13 | Hymenolobium | mesoamericanum | Fabaceae | 95.6 | 61.9 | 129.0 | 5.9 | 129.0 |
| 26 | 0.4 | 7 | Goethalsia | meiantha | Malvaceae | 63.6 | 60.9 | 74.7 | 1.9 | 74.7 |
| 27 | 0.4 | 6 | Otoba | novogranatensis | Myristicaceae | 69.8 | 61.9 | 88.9 | 4.2 | 88.9 |
| Rank | % | N | Genus | Species | Family | Diameter (cm) | | | | |
| | | | | | | Mean | Min | Max | SEM | Q05 |
| 28 | 0.4 | 6 | Spachea | correae | Malpighiaceae | 77.2 | 63.6 | 105.5 | 6.5 | 105.5 |
| 29 | 0.3 | 5 | Unidentified | Unidentified | Unidentified | 64.5 | 60.0 | 69.4 | 1.9 | 69.4 |
| 30 | 0.3 | 5 | Brosimum | lactescens | Moraceae | 67.2 | 63.7 | 78.9 | 2.9 | 78.9 |
| 31 | 0.3 | 5 | Pterocarpus | rohrii | Fabaceae | 72.7 | 60.2 | 90.8 | 5.0 | 90.8 |
| 32 | 0.2 | 4 | Chrysophyllum | colombianum | Sapotaceae | 73.2 | 66.7 | 83.3 | 3.6 | 83.3 |
| 33 | 0.2 | 4 | Ocotea | hartshorniana | Lauraceae | 72.6 | 64.8 | 89.1 | 5.6 | 89.1 |
| 34 | 0.2 | 4 | Luehea | seemannii | Malvaceae | 99.3 | 65.8 | 151.7 | 18.4 | 151.7 |
| 35 | 0.2 | 4 | Ficus | popenoei | Moraceae | 136.3 | 106.0 | 159.0 | 11.6 | 159.0 |
| 36 | 0.2 | 3 | Tabernaemontana | arborea | Apocynaceae | 61.3 | 60.2 | 62.5 | 0.7 | 62.5 |
| 37 | 0.2 | 3 | Simarouba | amara | Simaroubaceae | 63.7 | 62.0 | 66.6 | 1.5 | 66.6 |
| 38 | 0.2 | 3 | Tetragastris | panamensis | Burseraceae | 66.2 | 65.8 | 66.8 | 0.3 | 66.8 |
| 39 | 0.2 | 3 | Byrsonima | crassifolia | Malpighiaceae | 66.9 | 63.2 | 72.5 | 2.8 | 72.5 |
| 40 | 0.1 | 2 | Jacaranda | copaia | Bignoniaceae | 60.9 | 60.8 | 60.9 | 0.1 | 60.9 |
| 41 | 0.1 | 2 | Xylopia | sericophylla | Annonaceae | 62.2 | 61.9 | 62.5 | 0.3 | 62.5 |
| 42 | 0.1 | 2 | Inga | pezizifera | Fabaceae | 62.3 | 61.7 | 62.8 | 0.6 | 62.8 |
| 43 | 0.1 | 2 | Clethra | costaricensis | Clethraceae | 68.2 | 67.4 | 69.0 | 0.8 | 69.0 |
| 44 | 0.1 | 2 | Calophyllum | brasiliense | Calophyllaceae | 70.1 | 69.6 | 70.5 | 0.5 | 70.5 |

*(Continued)*

**Table 2.** (Continued)

| Rank | % | N | Genus | Species | Family | Mean | Min | Max | SEM | Q05 |
|------|---|---|-------|---------|--------|------|-----|-----|-----|-----|
| 45 | 0.1 | 2 | *Lacmellea* | *panamensis* | Apocynaceae | 69.7 | 61.4 | 78.0 | 8.3 | 78.0 |
| 46 | 0.1 | 2 | *Swartzia* | *nicaraguensis* | Fabaceae | 71.6 | 64.5 | 78.7 | 7.1 | 78.7 |
| 47 | 0.1 | 2 | *Handroanthus* | *chrysanthus* | Bignoniaceae | 70.8 | 61.4 | 80.1 | 9.4 | 80.1 |
| 48 | 0.1 | 2 | *Richeria* | *dressleri* | Phyllanthaceae | 74.2 | 65.1 | 83.3 | 9.1 | 83.3 |
| 49 | 0.1 | 2 | *Couepia* | *janzenii* | Chrysobalanaceae | 80.4 | 77.2 | 83.6 | 3.2 | 83.6 |
| 50 | 0.1 | 2 | *Ampelocera* | *macrocarpa* | Ulmaceae | 81.5 | 79.0 | 84.0 | 2.5 | 84.0 |
| 51 | 0.1 | 2 | *Coccoloba* | *tuerckheimii* | Polygonaceae | 72.1 | 60.0 | 84.2 | 12.1 | 84.2 |
| 52 | 0.1 | 2 | *Pterocarpus* | *hayesii* | Fabaceae | 91.8 | 72.1 | 111.5 | 19.7 | 111.5 |
| 53 | 0.1 | 1 | *Lonchocarpus* | *ferrugineus* | Fabaceae | 60.4 | 60.4 | 60.4 | - | 60.4 |
| 54 | 0.1 | 1 | *Abarema* | *adenophora* | Fabaceae | 60.6 | 60.6 | 60.6 | - | 60.6 |
| 55 | 0.1 | 1 | *Pourouma* | *bicolor* | Urticaceae | 61.4 | 61.4 | 61.4 | - | 61.4 |
| **Rank** | **%** | **N** | **Genus** | **Species** | **Family** | **Diameter (cm)** | | | | |
| | | | | | | **Mean** | **Min** | **Max** | **SEM** | **Q05** |
| 56 | 0.1 | 1 | *Genipa* | *americana* | Rubiaceae | 61.7 | 61.7 | 61.7 | - | 61.7 |
| 57 | 0.1 | 1 | *Pouteria* | *durlandii* | Sapotaceae | 61.7 | 61.7 | 61.7 | - | 61.7 |
| 58 | 0.1 | 1 | *Ocotea* | *macropoda* | Lauraceae | 62.5 | 62.5 | 62.5 | - | 62.5 |
| 59 | 0.1 | 1 | *Inga* | *leiocalycina* | Fabaceae | 62.6 | 62.6 | 62.6 | - | 62.6 |
| 60 | 0.1 | 1 | *Annona* | *amazonica* | Annonaceae | 63.0 | 63.0 | 63.0 | - | 63.0 |
| 61 | 0.1 | 1 | *Eschweilera* | *collinsii* | Lecythidaceae | 63.9 | 63.9 | 63.9 | - | 63.9 |
| 62 | 0.1 | 1 | *Pterocarpus* | *officinalis* | Fabaceae | 65.2 | 65.2 | 65.2 | - | 65.2 |
| 63 | 0.1 | 1 | *Pouteria* | *reticulata* | Sapotaceae | 65.3 | 65.3 | 65.3 | - | 65.3 |
| 64 | 0.1 | 1 | *Brosimum* | *guianense* | Moraceae | 69.8 | 69.8 | 69.8 | - | 69.8 |
| 65 | 0.1 | 1 | *Dalbergia* | *melanocardium* | Fabaceae | 73.4 | 73.4 | 73.4 | - | 73.4 |
| 66 | 0.1 | 1 | *Terminalia* | *bucidoides* | Combretaceae | 79.2 | 79.2 | 79.2 | - | 79.2 |
| 67 | 0.1 | 1 | *Dussia* | *cuscatlanica* | Fabaceae | 85.0 | 85.0 | 85.0 | - | 85.0 |
| 68 | 0.1 | 1 | *Sloanea* | *laevigata* | Elaeocarpaceae | 89.0 | 89.0 | 89.0 | - | 89.0 |
| 69 | 0.1 | 1 | *Tabebuia* | *rosea* | Bignoniaceae | 91.1 | 91.1 | 91.1 | - | 91.1 |
| 70 | 0.1 | 1 | *Pouteria* | *silvestris* | Sapotaceae | 94.5 | 94.5 | 94.5 | - | 94.5 |

Abundance, diversity and size statistics for species of large trees in old-growth tropical rain forest at the La Selva Biological Station, Costa Rica. All large trees (N = 1622) in 238 0.50 ha plots were mapped, measured and identified (see Methods). Taxonomic categories follow the La Selva Digital Flora. SEM = standard error of the mean, Q95 is the limit of the 95% quantile.

**Table 3. Large tree species diversity by size.**

| Lower diameter limit (cm) | N large trees | Total basal area m² | N species |
|---------------------------|---------------|----------------------|-----------|
| 60 | 1622 | 736.7 | 70 |
| 70 | 805 | 469.3 | 50 |
| 80 | 406 | 294.7 | 39 |
| 90 | 203 | 181.8 | 25 |
| 100 | 105 | 113.5 | 18 |
| 110 | 58 | 72.5 | 13 |
| 120 | 33 | 46.4 | 10 |
| 130 | 16 | 25.5 | 9 |

The number of individuals, species, and total basal area of 1622 large trees in 238 0.5 ha plots in old-growth tropical rain forest at the La Selva Biological Station, Costa Rica, in relation to different stem diameter limits. Each higher diameter class is a subset of the adjacent lower diameter class.

**Table 4. Large trees crown position and basal area.**

| Mean crown position | N All large trees | Sum of basal area | N *Pentaclethra* large trees | N other large tree species |
|---|---|---|---|---|
| <3 | 93 | 34.7 | 76 | 17 |
| 3/3.5 | 229 | 93.9 | 162 | 67 |
| 4/4.5 | 1091 | 478.6 | 586 | 505 |
| 5 | 209 | 129.5 | 7 | 202 |

Distribution of crown positions [25] and basal area for 1622 large trees (≥60 cm diameter) in 238 0.50 ha plots over an upland old-growth tropical rain forest landscape at the La Selva Biological Station, Costa Rica. Crown positions were evaluated by two observers and averaged. 5 = fully emergent with full vertical and lateral illumination, 4 = >90% of the crown with direct vertical illumination, 3 = 10–90% of the crown with vertical illumination, <3 = <10% of the crown with vertical illumination (combining several less-illuminated categories down to no direct vertical or lateral illumination).

relating the density of large trees to modelled slope and elevation had an $r^2$ of 0.36 (P<0.05). In general, large tree density was higher on flatter plots at lower elevations (plot mean elevation range 53–129 masl). The same elevation and slope model applied to the large tree plots showed a similar pattern ($r^2$ = 0.10, P<0.001, more large trees in the flatter and lower-elevation plots).

## Large tree distribution in relation to soil nutrients

There were numerous differences in stand structure for large trees and smaller stems between sites on different soils types with similar flat topographies (Table 5 column Soil Effect). In the long-term inventory plots on flat old alluvial soil large tree density was approximately twice as high as on flat residual soil sites. Plot-level large tree basal area and EAGB were also significantly higher on the old alluvial soils. Although stems ≥60 cm diameter had significantly larger mean diameters on the flat alluvial soils compared to residual soils, they were significantly

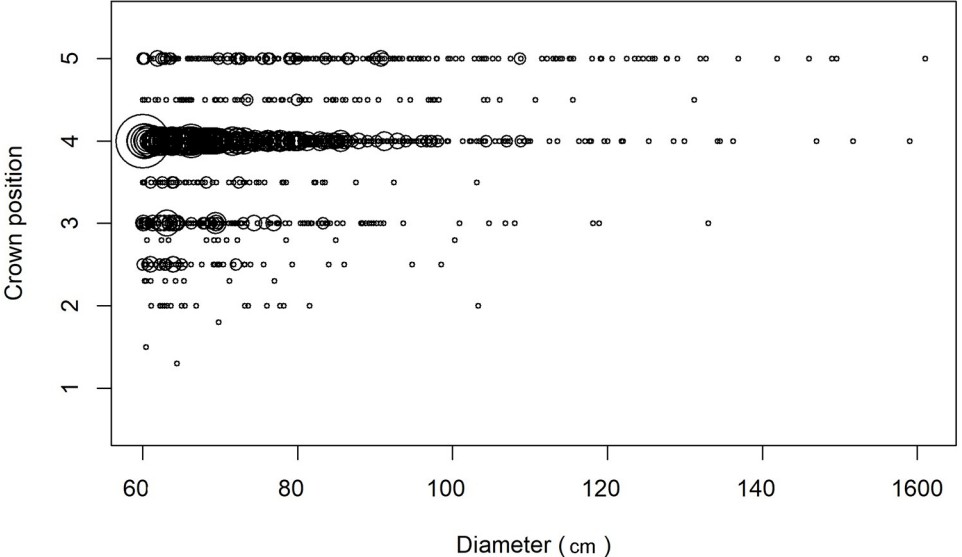

**Fig 2. Large tree crown position by trunk diameter.** Crown position, an index of exposure to illumination [25], of 1622 large trees in 238 0.5 ha plots across an upland old-growth tropical rain forest landscape at the La Selva Biological Station, Costa Rica. Crown position codes: 5 = fully emergent (exposed to both lateral and overhead light), 4 = ≥90% of crown exposed to vertical illumination; 3 = 10–90% vertical illumination; 2 = only lateral illumination, 1 = no direct illumination. Crown position was evaluated by two observers, and the average of the two estimates was calculated for each tree. Symbol size is proportional to the square root of the number of observations (range = 1–27).

**Table 5. Large tree abundance in relation to soil type.**

|  | Old alluvial | SEM | Flat residual | SEM | Steep residual | SEM | Soil effect | Slope effect |
|---|---|---|---|---|---|---|---|---|
| **Mean number large trees per plot** | 9.2 | 0.9 | 5.0 | 0.8 | 4.5 | 1.4 | ** | NSD |
| **Mean number smaller trees per plot** | 191.5 | 11.1 | 259.7 | 4.7 | 267.0 | 8.7 | *** | NSD |
| **Mean large tree stem diameter per plot (cm)** | 73.5 | 1.64 | 70.6. | 1.86 | 70.9 | 3.42 | NSD | NSD |
| **Mean smaller trees stem diameter per plot (cm)** | 20.6 | 0.33 | 19.0 | 0.29 | 19.6 | 0.21 | ** | NSD |
| **Mean total large trees BA per plot (m2)** | 4.0 | 0.5 | 2.0 | 0.3 | 1.9 | 0.6 | ** | NSD |
| **Mean total smaller trees BA per plot (m2)** | 8.3 | 0.3 | 9.3 | 0.3 | 10.3 | 0.4 | * | NSD |
| **Mean total large tree EAGB per plot (Mg)** | 33.6 | 4.2 | 16.3 | 2.6 | 15.1 | 5.0 | ** | NSD |
| **Mean total smaller trees EAGB per plot (Mg)** | 55.3 | 1.6 | 59.1 | 2.2 | 66.0 | 2.7 | NSD | NSD |

Density, size and contribution to plot basal area and EAGB of large trees and smaller stems (10 - <60 cm diameter) across the three dominant upland edaphic conditions in old-growth at La Selva, based on the data from 18 0.5 ha plots, 1997–2017. The metrics are means calculated over the six plots in each edaphic condition. Mean large tree diameter in plot P2 was based on only 16 years, because there were no large trees present for 5 years; all other plots are based on N = 21. SEM = standard error of the mean, BA = basal area, EAGB = estimated above-ground biomass, NSD = no significant difference. Tests for soil effects are based on ANOVA between the longs-term means from the six plots of flat old alluvial sites and the six flat residual soil plots (different soils, similar topographies). Tests for slope effects are based on ANOVA between the long-term means from the six plots on flat residual soil and the six plots on steep residual soil (similar soils, different topographies).

* P≥0.05,

** P≥0.01,

*** P≥0.001.

denser and had higher plot basal area on the residual soil plots. Across the 18 long-term inventory plots large tree density was significantly correlated with soil concentrations of P, K and Mg in the top 10 cm ($r_{(adj)}$ = 0.41, 0.52, -0.48 respectively, P≥0.05 for all; soil data in S4 Table). A 3-factor model relating large tree density to concentrations of P, K and Mg had an $r^2_{(adj)}$ of 0.41 (P<0.05).

In the 238 large tree plots, the density of large trees was also higher in plots that were principally or entirely on old alluvial soil compared to plots on residual soils (9.3 vs. 6.4 large trees/plot, S5 Table). We did not have plot-level nutrient analyses for the large tree plots and so could not examine the effects of specific nutrients on large tree density in this data set.

## Large tree productivity in relation to edaphic factors

There were no significant differences in mean diameter growth rates of individual large trees and smaller stems among the edaphic categories (Table 6). In contrast, plot-level large tree basal area and EAGB increments were higher on flat old alluvial sites and higher for smaller stems on flat residual soil sites. The differences in productivity were caused primarily by the significantly higher density of large trees on alluvial soils and of smaller stems on residual soils (Table 5). No productivity differences related to flat versus sloping sites on residual soils were found for either large trees or smaller stems.

## Large tree dynamics

We analyzed large tree death rates in the large trees- and long-term inventory plots over the time interval that the large tree plots were studied (2006–2016). Rates of annual mortality of the large trees were very similar between the CARBONO and large tree inventory plots (2.53% vs. 2.46% respectively, Table 7.) Over this decade rates of mortality in the CARBONO plots were not significantly different among stem diameter size categories ($X^2$ = 1.818, df = 5, NS).

Large tree mortality and recruitment did not vary with soil type (Table 8). For smaller stems mortality and recruitment rates were higher in flat residual soil plots than in old alluvial

**Table 6. Stem growth across edaphic categories.**

| Performance metric | Flat alluvial soil sites | | Flat residual soil sites | | Steep residual soil sites | | Effect | |
|---|---|---|---|---|---|---|---|---|
| | Mean | SEM | Mean | SEM | Mean | SEM | Soil | Slope |
| Annual large tree diameter growth (mm) | 4.52 | 0.39 | 3.60 | 0.74 | 3.38 | 0.52 | NSD | NSD |
| Annual diameter growth (mm) of smaller stems | 2.78 | 0.10 | 3.07 | 0.16 | 2.93 | 0.22 | NSD | NSD |
| Annual basal area increment per individual large tree (m$^2$) | 0.00512 | 0.00047 | 0.00400 | 0.00094 | 0.00371 | 0.00067 | NSD | NSD |
| Annual basal area increment per smaller individual (m$_2$) | 0.00116 | 0.00005 | 0.00104 | 0.00003 | 0.00106 | 0.00007 | NSD | NSD |
| Annual biomass increment per individual large trees (kg) | 45.1 | 4.2 | 35.2 | 8.3 | 32.6 | 5.9 | NSD | NSD |
| Annual biomass increment per smaller individual (kg) | 9.0 | 0.4 | 7.7 | 0.2 | 7.9 | 0.5 | * | NSD |
| Annual large tree total basal area increment per plot (m$^2$) | 0.047 | 0.008 | 0.017 | 0.003 | 0.019 | 0.010 | ** | NSD |
| Annual total basal area increment per plot of smaller stems (m$^2$) | 0.219 | 0.006 | 0.272 | 0.012 | 0.283 | 0.023 | ** | NSD |
| Annual large tree total EABG increment per plot (Mg) | 0.413 | 0.066 | 0.152 | 0.029 | 0.170 | 0.088 | ** | NSD |
| Annual total EABG increment per plot for smaller stems (Mg) | 1.694 | 0.041 | 2.008 | 2.008 | 2.118 | 0.171 | ** | NSD |

Growth of large trees and smaller stems across an edaphic gradient in old-growth tropical rain forest at the La Selva Biological Station, Costa Rica. Analyses were based on means of annual plot-level means or sums, 1997–2017, N = 6 plots per edaphic condition. Mean large tree diameter growth in plot P2 was based on only 14 years, because there were no large trees present for 6 years; all other plots are based on N = 20. Tests for soil effects contrast flat alluvial soil plots with flat residual soil plots, tests for slope effects contrast flat residual soil plots with steeply sloping residual soil plots. SEM = standard error of the mean

* P≥0.05,

** P≥0.01,

NSD no significant difference.

plots. For both large trees and smaller trees there were no significant differences in mortality between the flat and steeply-sloping plots on residual soils, but small stem recruitment was higher on flat residual-soil sites compared to steeply sloping ones (Table 8).

In the long-term inventory plots mortality of large trees accounted for an average of 22% of the total annual basal area loss from 1997–2017 (S6 Table). Large tree basal area loss ranged from 8% to 56% of total annual basal area lost in the plots. This was more than twice as variable as annual basal area loss by smaller stems (CV 79% v 31%), so that interannual variance in landscape-scale basal area loss was primarily driven by the death of large trees. There was no relation between total basal area lost in a year by large trees and by smaller stems ($r_{adj}^2$ = 0.02, P = 0.53, N = 20), suggesting different causative factors for mortality in the two groups.

**Table 7. Tree mortality.**

| | Large tree inventory plots | | | CARBONO plots 2006–2016 | | | |
|---|---|---|---|---|---|---|---|
| First census | N$_o$ | N 2016 | m | Diameter class cm | N$_0$ | N$_1$ | m |
| 2006 | 84 | 70 | 1.81 | 10.0–19.9 | 3016 | 2360 | 2.42 |
| 2007 | 137 | 114 | 2.02 | 20.0–29.9 | 680 | 524 | 2.57 |
| 2008 | 587 | 462 | 2.95 | 30.0–39.9 | 313 | 247 | 2.34 |
| 2009 | 522 | 436 | 2.54 | 40.0–49.9 | 202 | 153 | 2.74 |
| 2010 | 92 | 75 | 3.35 | 50.0–59.9 | 112 | 84 | 2.84 |
| 2011 | 200 | 185 | 1.55 | GTEq 60.0 | 99 | 78 | 2.36 |
| Average | | | 2.37 | | | | 2.54 |
| Weighted average | | | 2.53 | | | | 2.46 |

Rates of tree mortality at the La Selva Biological Station, Costa Rica. **m** is the exponential annual rate of mortality [48]. **N 2016** is number of large trees alive at the 2016 census for cohorts with different first census dates. The **weighted average** annual mortality is the sum of each first census sample size multiplied by the associated mortality rate, divided by the sum of sample sizes [49].

**Table 8. Mortality and recruitment by edaphic type.**

| Size Class | Old alluvial soils | | Residual soils flat | | Residual soils sloping | | Soil effect | Slope effect |
|---|---|---|---|---|---|---|---|---|
| | m | N1997 | m | N1997 | m | N1997 | | |
| Large trees | 1.86 | 48 | 3.11 | 32 | 3.20 | 23 | NSD | NSD |
| Smaller stems | 2.27 | 1183 | 2.65 | 1502 | 2.73 | 1633 | * | NSD |
| | r | N1997 | r | N1997 | r | N1997 | | |
| Large trees | 2.46 | 48 | 2.26 | 32 | 3.30 | 23 | NSD | NSD |
| Smaller stems | 1.41 | 1183 | 2.17 | 1502 | 1.60 | 1633 | *** | *** |

Annual tree mortality and recruitment by edaphic type and size class from 1997–2017 in 18 0.50 ha plots sited across an upland soil gradient in old-growth tropical wet forest at the La Selva Biological Station, Costa Rica. Calculation of exponential annual mortality (m) followed Sheil and May [48]. Soil effects were examined using contingency table analysis of deaths and survivals for all individuals first censused in 1997 and recensused in 2017 on flat alluvial soils and steeply sloping residual soils. Slope effects were examined by contrasting individuals on flat residual soils versus steeply sloping residuals soils. Annual recruitment (r) by soil type was calculated as post-1997 recruits present in 2017 compared to the initial size cohort in 1997; individuals that recruited after 1997 and that died before 2017 were not included [50]. Contingency analysis compared number of surviving recruits to total small stems present in 1997 as described above for mortality

* $P \geq 0.05$,

*** $P < 0.001$,

NSD No Significant Difference.

### Decadal trends in large tree density and dynamics

To analyze large tree density and dynamics over the two decades of this study we considered the entire upland old-growth landscape sampled by the 18 0.5 ha long-term inventory plots. Over the 20-year study period stem density (all stems $\geq 10$ cm diameter) showed no directional trend, ending at almost the same density as in the beginning of the study (S1 Fig). In contrast, mean stem diameter per plot increased steadily (S2 Fig), ending about 5% higher. As a result, mean plot basal area and EAGB were 8%-9% greater at the end of the study (S3 Fig), with an apparent leveling off in the last six years. An increase in stand basal area does not necessarily lead to an increase in stand biomass if species composition is changing to species with lower wood density [51,52]. Because EAGB-weighted wood density for trees and palms did not change over the 20 years (0.491 g/cc$^3$ 1997 and 2017), the increase in basal area led to actual increases in stand biomass.

In contrast to total stem density, large tree density increased 27% over the 20 years (Fig 3). Mean large tree stem diameter decreased from 73.7 cm to 71.1 cm (S4 Fig), while mean plot large tree basal area increased 22% (S5 Fig).

The increase in large tree plot basal area began in the mid-2000s, and coincided with a 12-yr period where total recruits to large tree status (those newly reaching $\geq 60$ cm diameter) outnumbered total large tree deaths by 59 to 31 (Fig 4). In 1997–1998 a strong El Niño event caused record high temperatures and record low dry season rainfall at La Selva [53]. After this event there was a sharp increase in diameter growth rates of both large and small stems, but subsequently diameter growth rates and total landscape basal area addition showed no directional trend (Fig 5, S6 Fig). The preponderance of large tree recruits over deaths in the last 13 years was not due to increases in diameter growth rates of individuals in the immediate pre-large tree size class (50.0–59.9 cm diameter) (S7 Fig).

## Discussion

### Decadal trends in large tree density

Any study of large trees, and of any particular tropical forest landscape, must deal with the issue of a limited period of observation relative to much longer tree lifespans and potential

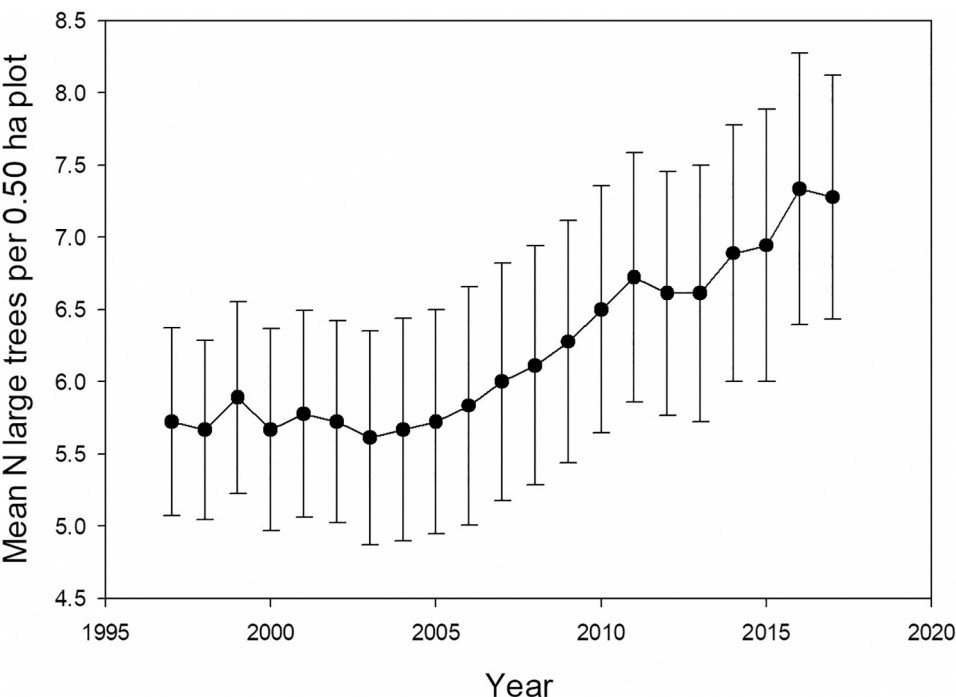

**Fig 3. Large tree density.** The mean number of large trees per plot in 18 0.50 ha long-term inventory plots from 1997–2017, ± 1 SEM.

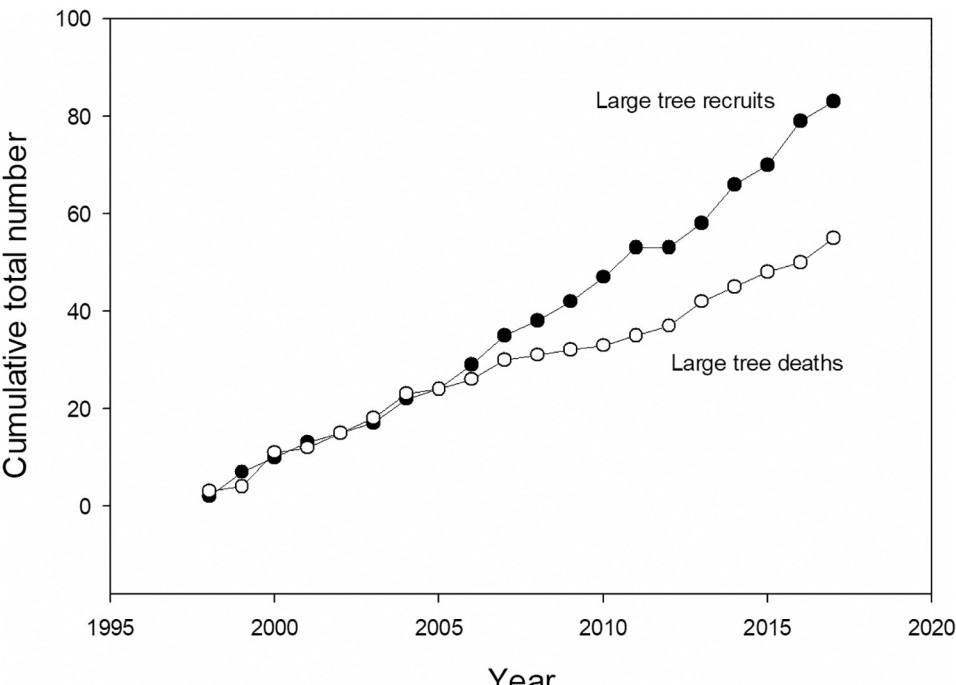

**Fig 4. Dynamics of large trees.** The cumulative number of large tree recruits and deaths for 18 0.50 ha CARBONO plots.

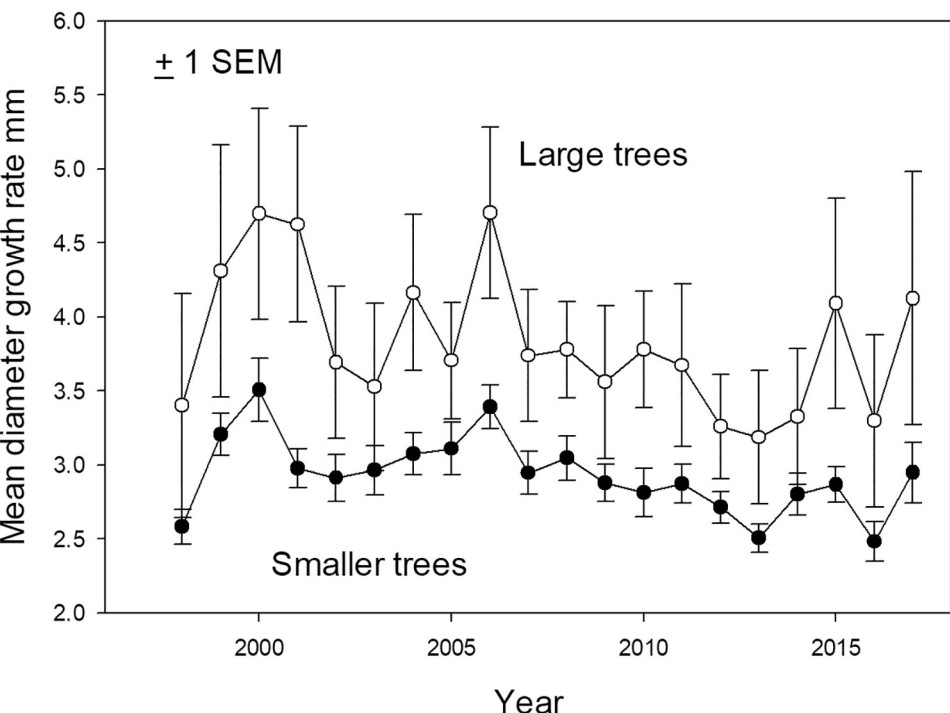

**Fig 5. Diameter growth.** Mean annual diameter growth (mm) in 18 0.50 ha plots 1997–2017 for all large trees (white symbols) and smaller stems (black symbols).

legacies of disturbance events prior to the study period [54]. Individual tropical forest large trees can live several centuries [55], while most tropical ecological studies span a few decades at most. In the case of the landscape studied here, the initial year of the intensive forest inventory plot coincided with a major El Niño event [16,53]. The effects of that event on forest structure and tree dynamics were substantial. However gross descriptors of forest structure and process returned to presumed baseline levels within a few years [16,53].

To understand large tree dynamics over this old-growth landscape over the last 20 years it is useful to contrast their performance and effects on forest structure with those of the smaller stems, which account for 97% stems and 73% of stand EAGB (Table 1). Over the last two decades total landscape stem density was impressively stable (S1 Fig). The total number of deaths was almost exactly offset by incoming recruits (2095 vs 2100, Fig 6). Over the same period average stem diameter increased (S2 Fig), so that mean plot basal area increased (S3 Fig). Increasing stand basal area has also been reported from other tropical rain forests, and rising global $CO_2$ concentrations have been hypothesized as a potential driver of this change [56]. Several studies [17,18, 57] now have reported increasing stand EAGB and rates of tree growth, mortality, and recruitment (the "Bigger and Faster Hypothesis" or B&FH$_0$ [16]).

While the old-growth landscape studied here did show increasing basal area over a two-decade interval (S3 Fig), trends in key demographic traits differed from those predicted by the B&FH$_0$. There was no increase in diameter growth rates over time (Fig 5). Instead of the increasing stand dynamics predicted by the B&FH$_o$, stand-level mortality and recruitment declined over the two-decade study period (S8 and S9 Figs). As a result, turnover (the average of mortality and recruitment) declined over the course of this study (Fig 7). Lower turnover leads to longer average tree lifetimes, or more time for trees to grow before death. Higher basal area was a demographic consequence of decreasing stand dynamism over the last two decades.

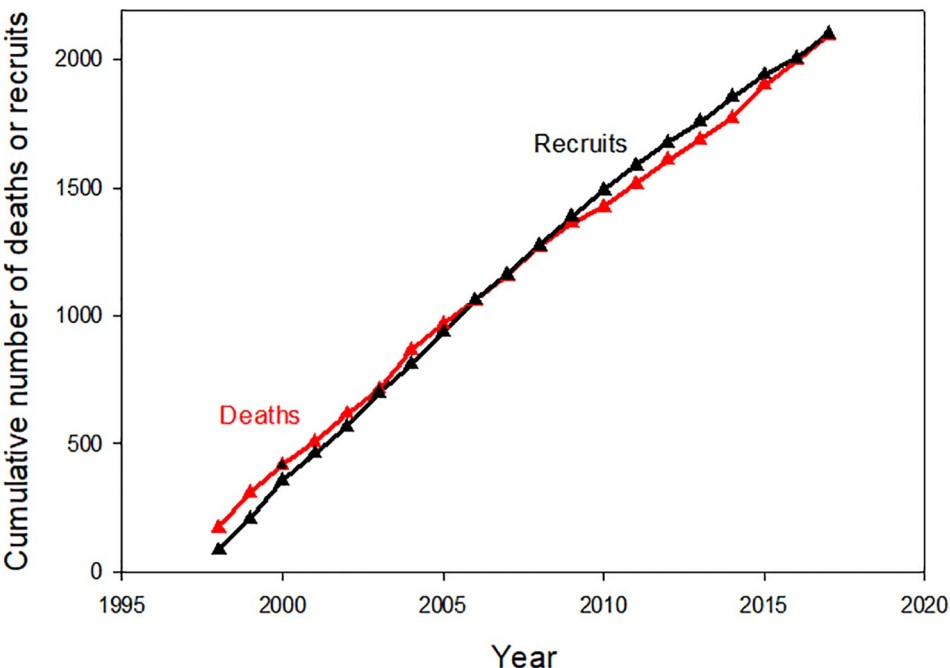

**Fig 6. Stem dynamics.** The cumulative total number of deaths (red line) and recruits (black line) for all stems ≥10 diameter in 18 0.50 ha annually-censused long-term inventory plots from 1997–2017 at the La Selva Biological Station, Costa Rica.

Large tree dynamics differed from those of the smaller stems over this landscape. In contrast to the landscape-scale stability of total stem density (S1 Fig), large tree density rose considerably from 1997 to 2017 (Fig 3), and mean large tree stem diameter decreased (S4 Fig). The increase in large tree stem density offset the decrease in mean stem diameter, so large tree basal area per plot increased (S5 Fig). The net result of increasing basal area is the same for large tree and smaller stems, but the underlying demographic processes are completely different. Over two decades ago Clark and Clark [36] suggested that, based on growth and death rates of a large marked sample of large trees at La Selva, large tree density must have been increasing at the landscape scale. The results from long-term inventory plots presented here confirm that inference.

The results for both large tree and smaller stems over the last two decades are consistent with a landscape recovering from past disturbance. It is clear that the strong 1997 El Niño event had significant impacts on forest structure and dynamics at La Selva (Fig 7, S3, S8 and S9 Figs, [16,53]). We do not have annual-scale data on forest dynamics prior to 1997. However, the two decades prior to our study was a period of unusually intense El Niño activity [58]. From 1895–2015 the 24 3-month periods of highest of El Niño indices includes only 6 different years of the 120 possible; two of these occurred during this study (1997, 1998), and the other four occurred within the preceding 15 years (1982, 1983, 1987, 1992). Given the magnitude of disturbance associated with the 1997 El Niño, it is possible that other major El Niño events during the two decades prior to 1997 caused similar impacts on La Selva. S1 Fig shows that total stand density recovers fairly quickly from natural disturbance, while Fig 7 and S2 Fig suggest a forest that is increasing in average tree age and size. In contrast, the large tree data suggest that large tree numbers have yet to recover from past disturbance in this forest. The

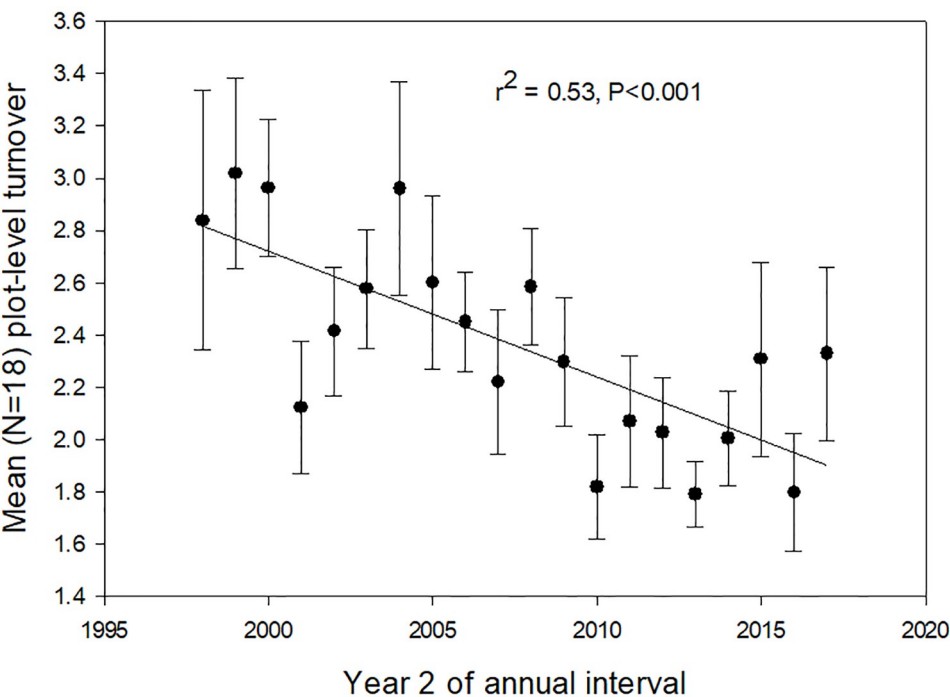

**Fig 7. Turnover.** Mean annual turnover (the average of annual recruitment and annual mortality) for all stems $\geq 10$ cm diameter in 18 0.5-ha plots in old-growth tropical rain forest, La Selva Biological Station, Costa Rica, 1997–2017. Error bars are ±1 SEM.

decreasing average size (S4 Fig) and excess of recruits over deaths (Fig 4) is a pattern consistent with recovery in the large tree size class in the last two decades.

In summary, the unusually detailed data from this study, for both large tree and smaller stems, are consistent with a landscape recovering from disturbance. While increasing plot basal area is also consistent with the B&FH$_o$, the demographic data (total stem density stable, decreasing turnover, no directional trend in growth rates or productivity) suggest that changes in forest structure and dynamics in the last two decades over this landscape have been driven primarily by local disturbance and recovery, and are not principally related to an accelerating global driver like increasing atmospheric $CO_2$ levels.

To our knowledge there is no other tropical forest landscape with the intensity of forest process sampling described in this paper, i.e. stratified random replicated forest inventory plots measured annually for 20 years. For this landscape the available evidence for both smaller trees and large trees points to a disturbance-recovery explanation for increasing stand basal area. Pantropically, most of the tropical rain forest sites that provided data for the Bigger and Faster Hypothesis were established between 1960 and 2000 (cf., median plot establishment date 1986, 1968–2005, N = 79 [59]; median 1991, 1972–2005, N = 137 [9]). The 36 strongest 3-month average El Niño events from 1895–2015 all occurred in this period (8 different years [58]). It is possible that many or most of the plots used as evidence for the B&FH$_O$ have also experienced major disturbance related to the unique history of strong El Niño events in recent decades, and also perhaps in the decades prior to plot establishment. This hypothesis is consistent with the results of Aleixo et al [13], who showed that tree mortality in the Central Amazon was strongly related to ENSO events. Although we have here focused on El Niño-related disturbance, there are also other disturbances that could produce similar effect (e.g. North Atlantic Oscillation anomaly [13], regional droughts [13,60], local storms [13, 61]). As more data

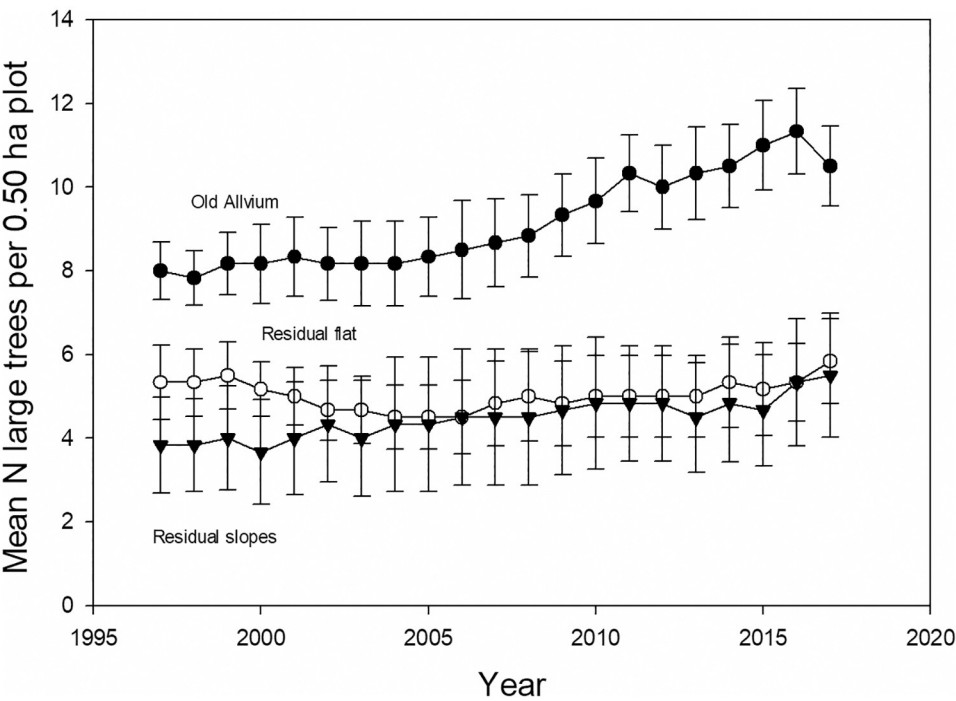

**Fig 8. Large tree density by soil type.** Mean number of large trees in 18 0.50 ha forest inventory plots in each of three dominant edaphic conditions in the old-growth tropical rain forest at the La Selva Biological Station, Costa Rica. N = 6 0.5-ha plots per treatment. Error bars are ± 1 SEM.

accumulate from tropical forest sites pantropically, it will be useful to determine if the size-related demographic trends found over the old-growth landscape at La Selva (smaller trees density stable, mean diameter increasing vs. larger tree density increasing, mean diameter decreasing) also occur in other tropical forests. If so, it may be necessary to reassess recent trends in tropical forest dynamics and to place a greater emphasis on decadal disturbance and recovery patterns and processes [54].

## Intra-landscape variation in large tree distribution and dynamics

There were substantial differences in large tree density and dynamics across the edaphic gradients of the old-growth landscape. Large tree stem number and basal area on residual soil plots were about half of those on old alluvial soil plots (Table 5). Large tree death rates were 70% higher on residual soils although the difference was not statistically significant (Table 8). The differences in large tree density among plots on different soil types increased over time (Fig 8). The plots on old alluvial soil also had higher soil concentrations of P and K (data in S4 Table), so the large tree patterns were correlated with soil nutrient levels.

The most direct expected effects of increasing soil nutrients on large tree performance would be increased growth rates on higher-nutrient sites. A plausible consequence of higher diameter growth rates would be higher rates of recruitment of smaller stems to large tree status. In fact, there were no significant differences in diameter growth rates of either large trees or smaller stems across soils types (Table 6 column Soil Effect) and the rates of large tree recruitment were not different among edaphic types (Table 8). There was therefore no clear evidence for direct nutrient effects as the principal factor accounting for the large intra-landscape differences in large tree density and dynamics.

In contrast, there is evidence that these within-landscape differences in forest structure and dynamics were due to spatially-localized disturbances. The largest disturbances over the study period observed were concentrated in residual soil areas. For the entire study period the median number of dead individuals per plot per annual census was 5 (N = 396 plot-censuses). For the largest 3% of these events however the median number of dead individuals was 21 (range 16–40, N = 12), and these all occurred on residual-soil plots. In addition, plots on residual soil were more dynamic than plots on old alluvial soil over these two decades, with significantly higher rates of mortality and recruitment of smaller individuals (Table 8).

As argued above, it is also likely that a series of disturbances linked to large El Niño effects affected this landscape in the two decades prior to our study. We have no way of knowing if disturbances prior to our study were also concentrated on the residual soil areas but it is a possibility.

Overall, our conclusion at this point is that spatially-localized disturbances, particularly the larger disturbances, have been a major factor in causing and/or sustaining the observed differences in large tree dynamics and distribution across the old-growth landscape. The potential impacts of such a spatially-concentrated disturbance were demonstrated by a powerful storm that hit La Selva on 19 May 2018. Post-storm impact studies are currently underway, but the effects on plots in the storm's path can be inferred from 2018 post-storm census data. Fourteen plots appeared largely unaffected by the storm. In these plots total stem number decreased 0.7% from 2017 to 2018. For the four NW plots in the storm's path however, total stem number decreased by 17.0%. This example, and the 12 largest disturbance events discussed above, indicate that localized disturbances large enough to cause significant structural and demographic effects on 0.50 ha stands were moderately common events on this landscape over these two decades. Chambers et al. [54] suggested that for Amazonian forests a plot size of 10 ha is necessary to "maximize detection of temporal trends" and account for rare disturbance events. Our results suggest that an alternative approach, ie replicated smaller plots, is a more informative approach, since it can explicitly incorporate sampling across mesoscale ecological gradients and also generate field-measured metrics of variance in key forest properties.

Today the La Selva old-growth tropical forest is the only site that we are aware of that has been assayed with a landscape-scale plot network with annual censuses over two decades. It is therefore currently not possible to say how typical or atypical the level of disturbances documented in this study is for tropical rain forests in general, or how representative these two decades are of century-scale disturbance regimes. There are however two studies that have reached conclusions similar to ours. Ruisthauser et al. [62] censused 6 6.25-ha plots biannually for 16 years and concluded "biomass net changes were mainly driven by large and unpredictable losses, whereas gains remained nearly constant over time." Murphy et al [37] censused 20 0.50 ha plots over 40 years. Plot biomass did not increase over this interval and stem density decreased. They conclude that in the Australian rain forests they studied "are either not increasing in productivity in response to global change, or cyclones and other regional and local mechanisms of change mask any evidence of larger-scale patterns."

The research reported here revealed significant intra-landscape variation in large tree density in an old growth tropical rain forest, as well as similar variation in disturbance history and its effects. With the increasing power and plunging costs of remotely-sensed data, it is increasingly feasible to assess other tropical landscapes to determine if similar intra-landscape variance in stand structure and tree demography is common. Determining the footprints of historical disturbance on large trees demography will be more difficult as a series of repeat censuses is required, but such data will become increasingly common with repeated sampling via remote sensing [15,26,27,63,64,65]. Future research could profitably focus on those study sites where existing ground plots can provide information on the smaller stems to complement and

inform analyses of large tree distribution and performance. Given the importance of large trees to tropical forest carbon cycling and forest structure and dynamics, and the likely vulnerability of large trees to changing global and local climate effects, comparative research on large tree distribution and dynamics is urgently needed.

## The concept of large trees: A practically useful classification with a complex biological foundation

The majority of the results presented in this paper are based on the classification "large trees" using an admittedly-arbitrary size limit. By using a strict size criterion, we and others using this approach specifically ignored species-level biological traits. The utility of using large trees as a practical method for understanding many aspects of tropical forest ecology has been amply demonstrated [1,2,11,14,31,64].

Are there biological traits other than maximum size shared by species that attain large tree status? At least over the old-growth tropical rain forest landscape and species pool that we studied, the answer is a qualified "No". While as a class large tree mortality was not different from smaller stems in the long-term inventory plots (Table 7), at the species level there were marked interspecific differences. Within the large tree plots, annual mortality rates of the 15 most common species varied by two orders of magnitude (0.04–9.59% yr$^{-1}$ S7 Table). We found that species-level mortality was significantly lower for species with larger 95$^{th}$ percentile, maximum and mean diameters (95$^{th}$ percentile diameter x weighted mean mortality $r_{(adj)}^2$ = 0.30, P<0.05, N = 15 species). Similarly, Thomas et al. [27] found that taller individuals at La Selva had lower death rates than shorter individuals. For species that ever had an individual that attained large tree status in the long-term inventory plots ("large tree species", N = 32), mean diameter growth rates varied by an order of magnitude (all annual diameter measurements of any size individual from 1997–2017, 1.3–16.2 mm/year). These large tree species also spanned two orders of magnitude in the percentage of the diameter measurements that occurred on large tree-sized stems (0.4 to 54.0%).

The striking interspecific variation we documented among large tree species was in part due to the arbitrary size limit used to define large trees, and partially due to the arbitrary classification of species as "large tree species" based on as few as one occurrence of a large tree-sized individual in that species' sample. Going forward, it is certainly possible to investigate other arbitrary size limits and species classification alternatives. We believe a more useful course is to acknowledge the biological diversity contained within any large tree classification, while also recognizing the practical utility of applying an arbitrary size limit for large trees to facilitate the study of many aspects of tropical forest ecology at large spatial scales. For investigating species-level traits associated with demographic traits (such as maximum size and death rates), a regression analysis that specifically incorporates a continuum of responses is a statistically more powerful and biologically more realistic approach than the fixed size limit used in stand-based large tree studies.

## Supporting information

**S1 Table. CARBONO plot Inventory data 1997_2017.**
(CSV)

**S2 Table. CARBONO plot dynamics data 1997–2017.**
(CSV)

**S3 Table. Large tree plots individual tree data.**
(CSV)

**S4 Table. Soil nutrient data for 0–10 cm La Selva Biological Station.**
(CSV)

**S5 Table. Large tree density and summed basal area in 238 0.50 ha plots.**
(DOCX)

**S6 Table. Total basal area loss by large trees and smaller stems.**
(DOCX)

**S7 Table. Large tree species-level annual mortality rates 2006–2016.**
(DOCX)

**S1 Fig. Long-term plots stem density 1997–2017.**
(TIF)

**S2 Fig. Mean stem diameter in 18 0.50 ha inventory plots 1997–2017.**
(TIF)

**S3 Fig. Mean annual plot basal area (A) and EAGB (B) in 18 0.50 ha plots from 1997–2017.**
(TIF)

**S4 Fig. Mean large tree stem diameter in 18 0.50 ha inventory plots from 1997–2017.**
(TIF)

**S5 Fig. Mean large tree plot-level basal area in 18 0.50 ha inventory plots from 1997–2017.**
(TIF)

**S6 Fig. Mean annual plot-level basal area production by large trees (A) and smaller stems (B) from 1997–2017.**
(TIF)

**S7 Fig. Mean diameter growth all stems 50–60 cm diameter from 1997–2017 in 18 0.50 ha inventory plots.**
(TIF)

**S8 Fig. Mean plot-level mortality 1997–2017.**
(TIF)

**S9 Fig. Mean plot-level recruitment 1997–2017.**
(TIF)

## Acknowledgments

We thank our Costa Rican field assistants Leonel Campos, William Miranda, Marcos Molina, Gilberth Hurtado, Juan Gabriel Huertas and Wagner Correa, for their hard work and dedication to quality control in the field and in the laboratory work. We are very grateful to the staff of the La Selva Biological Station for their superb logistic and administrative support. Ricardo Sandi provided helpful assistance with La Selva's GIS databases, and Wagner Lopez created Fig 1.

## Author Contributions

**Conceptualization:** David B. Clark.

**Formal analysis:** David B. Clark, Antonio Ferraz.

**Funding acquisition:** Deborah A. Clark, Susan G. Letcher, Sassan Saatchi.

**Methodology:** Antonio Ferraz.

**Project administration:** David B. Clark, Deborah A. Clark.

**Supervision:** Deborah A. Clark.

**Writing – original draft:** David B. Clark.

**Writing – review & editing:** David B. Clark, Antonio Ferraz, Deborah A. Clark, James R. Kellner, Susan G. Letcher, Sassan Saatchi.

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
