## [Decision Letter · Decision Letter 0]

28 Aug 2019

PONE-D-19-18209

Diversity, distribution and dynamics of Very Large Trees across an old-growth lowland tropical rain forest landscape

PLOS ONE

Dear Dr. Clark,

Thank you for submitting your manuscript to PLOS ONE. After careful consideration, we feel that it has merit but does not fully meet PLOS ONE’s publication criteria as it currently stands. Therefore, we invite you to submit a revised version of the manuscript that addresses the points raised during the review process.

We would appreciate receiving your revised manuscript by Oct 12 2019 11:59PM. To enhance the reproducibility of your results, we recommend that if applicable you deposit your laboratory protocols in protocols.io, where a protocol can be assigned its own identifier (DOI) such that it can be cited independently in the future. For instructions see: http://journals.plos.org/plosone/s/submission-guidelines#loc-laboratory-protocols

We look forward to receiving your revised manuscript.

Kind regards,

RunGuo Zang

Academic Editor

PLOS ONE

Journal Requirements:

2. In your Methods section, please provide additional location information of the study area, including geographic coordinates for the data set if available.

Additional Editor Comments (if provided):

The two referees are both specialists in ecology of very large trees.They are positive to your manuscript,but they want your reports to be more clearer and give more discussions on the implications of your work.

Reviewers' comments:

Reviewer's Responses to Questions

**Comments to the Author**

1. Is the manuscript technically sound, and do the data support the conclusions?

Reviewer #1: Yes

Reviewer #2: Yes

2. Has the statistical analysis been performed appropriately and rigorously? 

Reviewer #1: Yes

Reviewer #2: Yes

3. Have the authors made all data underlying the findings in their manuscript fully available?

Reviewer #1: Yes

Reviewer #2: Yes

4. Is the manuscript presented in an intelligible fashion and written in standard English?

Reviewer #1: Yes

Reviewer #2: Yes

5. Review Comments to the Author

Reviewer #1: This is an important contribution to the existing literature on tropical forest dynamics and it fits perfectly in the ongoing debate whether tropical forests have been gaining biomass over the last decades or not, and if so, through which mechanism and process. I have very few comments:

1. I am curious whether it is possible to detect any directional pattern in species composition for the large trees over time, i.e. are the species that drop out different from the ones that come in? If this is the case it may affect the biomass calculations because the Brown equation that was used does not take any species specific traits (like wood density) into account. Directional change in species composition may make this equation less suitable for long term biomass monitoring than the more recent pan-tropical equations that do include such traits. Additionally, it may tell us something about the changes that are happening in the forest.

2. Line 202 states that of the 246 new species in the plots, 32 seem reached 60 cm dbh. Does this mean that they grew extremely fast? Or do you mean that those 32 species were observed to be able to reach 60 cm or more based on the larger sample of plots that you had? Please formulate this more clearly.

3. Line 218: I knew that Amercan trees grew less big than Asian trees, but that only 18 of the 1662 reached 100 cm still comes as a surprise to me.... In our Asian plots trees easily reach that size.... This remains an interesting puzzle to be solved.... Or would these forests still be recovering from disturbances that happened hundreds of years ago?

4. Is the higher large tree mortality on residual soils (compared to the alluvial soils) perhaps related to the soil water content? Drought mortality of large trees in Asia seems strongly correlated with topographic position, with higher mortality at slopes and ridges, and lowest mortality in valleys (higher soil water content).

Reviewer #2: Generally a well written and analysed manuscript. The manuscript adds to the growing literature on the importance of large trees globally. The major issue is that the discussion needs to be written with more clarity. To the reader, the main results presented are 1) general reporting of VLT abundance, diversity etc, 2) Effect of topography and soil nutrients, 3) Changes in VLT over time. These are fantastic results but need to be discussed more succinctly with some ecological implications and comparison to other studies. As both 2 and 3 relate predominantly to past disturbance, I suggest first discussing both and then relating each to past disturbance in a single paragraph. VLT crown exposure to light is included under a major results heading but not discussed. I suggest including one or two sentences in another part of the results or leaving it out completely. The use of remote sensing seems to be a major conclusion of your manuscript but is not part of your aims or results. I suggest leaving this out as it is covered numerous times in the literature. Instead, I would like to see some implications of your work (maybe carbon storage, projected recruitment and mortality) that you have touched on in the very last sentence. The references to future research add nothing to the manuscript.

Minor comments

Line 31 consider inserting ‘and contribution to forest structure and biomass’.

Consider simply using the term ‘large trees’ or ‘large diameter trees’ as is used in other publications, especially if using >60cm.

Consider ‘tropical rainforests’ (TRF) and ‘biomass’ (EAGB) as the multiple initialisms become hard to read.

Line 84I’m not sure what ‘individual conditions’ are.

Line 121delete the word ‘issues’ and check remainder of manuscript.

There is a mix of cm and mm diameter in the manuscript.

Line 340delete among

Line 415for global readers, indicate what the implications of an El Nino year are for your study area.

For the topic of long term dynamics following disturbance I suggest reading Murphy HT, et al. (2014) No evidence for long-term increases in biomass and stem density in the tropical rain forests of Australia. Journal of Ecology. 2013;101(6):1589-97.

For similar work on large diameter trees in Australian rainforests I suggest reading Bradford et al. (2019) The importance of large diameter trees in the wet tropical rainforests of Australia. PLoS ONE 14(5).

6. PLOS authors have the option to publish the peer review history of their article (what does this mean?). If published, this will include your full peer review and any attached files.

Reviewer #1: No

Reviewer #2: No

---

## [Author Response · Author response to Decision Letter 0]

21 Oct 2019

We greatly appreciate the constructive and useful questions and suggestions from the reviewers. We believe the revised manuscript incorporating their input is a significantly clearer and more readable presentation. We responded to all of their input in our responses below. We show the reviewers’ comments in italics, and our answers follow each comment in regular type.

Reviewer #1: This is an important contribution to the existing literature on tropical forest dynamics and it fits perfectly in the ongoing debate whether tropical forests have been gaining biomass over the last decades or not, and if so, through which mechanism and process. I have very few comments:

1. I am curious whether it is possible to detect any directional pattern in species composition for the large trees over time, i.e. are the species that drop out different from the ones that come in? If this is the case it may affect the biomass calculations because the Brown equation that was used does not take any species specific traits (like wood density) into account. Directional change in species composition may make this equation less suitable for long term biomass monitoring than the more recent pan-tropical equations that do include such traits. Additionally, it may tell us something about the changes that are happening in the forest.

 The reviewer raises an interesting point that we had not previously considered. In fact the basal area-weighted and biomass-weighted wood density did not change at all during the 20-year study period, so basal area increases did in fact lead to biomass increases. We included a new analysis and some discussion to document this point (lines 400-404).

2. Line 202 states that of the 246 new species in the plots, 32 seem reached 60 cm dbh. Does this mean that they grew extremely fast? Or do you mean that those 32 species were observed to be able to reach 60 cm or more based on the larger sample of plots that you had? Please formulate this more clearly.

 Line 202 said “Over these two decades 246 species of trees occurred in the plots.” There is no mention of “new species” so we’re unclear on the reviewer’s confusion. We revised the text to “). Over these two decades 241 species of trees occurred in the plots”

3. Line 218: I knew that Amercan trees grew less big than Asian trees, but that only 18 of the 1662 reached 100 cm still comes as a surprise to me.... In our Asian plots trees easily reach that size. This remains an interesting puzzle to be solved…..Or would these forests still be recovering from disturbances that happened hundreds of years ago?

 The text states that “18 species reached 100 cm diameter”, so the relevant comparison is 18 of the 70 VLT species (line 216) reached 100 cm. The number of VLT individuals sampled that reached 100 cm is given in Table 3 (105 of 1622). (emphasis added).

4. Is the higher large tree mortality on residual soils (compared to the alluvial soils) perhaps related to the soil water content? Drought mortality of large trees in Asia seems strongly correlated with topographic position, with higher mortality at slopes and ridges, and lowest mortality in valleys (higher soil water content).

 The apparent higher mortality of larger trees on residual soils (Soil Effect, Table 8) was in fact not statistically significant (per Table 8), so we prefer not to speculate on this non-significant result.

Reviewer #2: Generally a well written and analysed manuscript. The manuscript adds to the growing literature on the importance of large trees globally. The major issue is that the discussion needs to be written with more clarity. To the reader, the main results presented are 1) general reporting of VLT abundance, diversity etc, 2) Effect of topography and soil nutrients, 3) Changes in VLT over time. These are fantastic results but need to be discussed more succinctly with some ecological implications and comparison to other studies. As both 2 and 3 relate predominantly to past disturbance, I suggest first discussing both and then relating each to past disturbance in a single paragraph. 

 As suggested by the reviewer, the Discussion begins with the discussion of large tree demography at the landscape and intra-landscape scales (the reviewer’s points 2 and 3), and then deals with his point 1 in the discussion of the implications of classification systems for large trees. We shortened the Discussion by removing lines 595-604 of the original manuscript.

VLT crown exposure to light is included under a major results heading but not discussed. I suggest including one or two sentences in another part of the results or leaving it out completely. 

 We added wording to the Abstract to highlight the crown condition results (lines 38-39). These results are discussed in lines 240-252.

The use of remote sensing seems to be a major conclusion of your manuscript but is not part of your aims or results. I suggest leaving this out as it is covered numerous times in the literature. Instead, I would like to see some implications of your work (maybe carbon storage, projected recruitment and mortality) that you have touched on in the very last sentence. 

 In response to the reviewer’s concern we deleted the entire paragraph on the evolution of remoted sensing research on VLTs (lines 595-604 in the originally-submitted manuscript).

The references to future research add nothing to the manuscript.

 In response to the reviewer’s concern we deleted the entire Conclusions section. We replaced some of the text with the Intralandscape section with some of the text from the Conclusions, overall leading to significant shortening of the text and deleting some concept repetition. 

Minor comments

Line 31 consider inserting ‘and contribution to forest structure and biomass’.

 Text revised to “their contribution to forest structure and dynamics.” (line 31)

Consider simply using the term ‘large trees’ or ‘large diameter trees’ as is used in other publications, especially if using >60cm.

 VLT changed to “large trees” throughout.

Consider ‘tropical rainforests’ (TRF) and ‘biomass’ (EAGB) as the multiple initialisms become hard to read. 

 TRF changed to tropical rain forest or tropical forests throughout. 

 EAGB is the standard term for Estimated Above-Ground Biomass and there is no convenient alternative.

Line 84 I’m not sure what ‘individual conditions’ are.

 Text changed to “individual crown conditions”

Line 121 delete the word ‘issues’ and check remainder of manuscript. 

 Text changed to “biodiversity and distribution patterns”.

There is a mix of cm and mm diameter in the manuscript.

 Tropical tree size data are typically presented in cm. We modified the manuscript to follow this convention (cf Tables 1, 2, 3, 5, 7). Diameter growth rates are most commonly reported in mm so we retained mm for growth data.

Line 340 delete among

 “Among” deleted as suggested

Line 415for global readers, indicate what the implications of an El Nino year are for your study area.

 We agreed with reviewer and added text on lines 414-415 explaining that the 1997-98 large El Niño event was characterized by record high temperatures and low dry season rainfall. 

For the topic of long term dynamics following disturbance I suggest reading Murphy HT, et al. (2014) No evidence for long-term increases in biomass and stem density in the tropical rain forests of Australia. Journal of Ecology.2013;101(6):1589-97.

 We thank the review for noting our oversight, we agree that this paper should have been discussed. We added the reference and discuss its findings in relation to our own (lines 580-584).

For similar work on large diameter trees in Australian rainforests I suggest reading Bradford et al. (2019) The importance of large diameter trees in the wet tropical rainforests of Australia. PLoS ONE 14(5).

 We agree that this recent paper is highly relevant and included citation to this work (L53).

Overall, we added a total of ten additional references to better document the text and also incorporate very recent publications.

---

## [Editor Report · Decision Letter 1]

24 Oct 2019

­Diversity, distribution and dynamics of large trees across an old-growth lowland tropical rain forest landscape

PONE-D-19-18209R1

Dear Dr. Clark,

We are pleased to inform you that your manuscript has been judged scientifically suitable for publication and will be formally accepted for publication once it complies with all outstanding technical requirements.

With kind regards,

RunGuo Zang

Academic Editor

PLOS ONE

Additional Editor Comments (optional):

Accept
---

## [Editor Report · Acceptance letter]

30 Oct 2019

PONE-D-19-18209R1 

­­Diversity, distribution and dynamics of large trees across an old-growth lowland tropical rain forest landscape 

Dear Dr. Clark:

I am pleased to inform you that your manuscript has been deemed suitable for publication in PLOS ONE. Congratulations! Your manuscript is now with our production department. 

With kind regards,

on behalf of

Professor RunGuo Zang 

Academic Editor

PLOS ONE